# Puerarin Alleviates H_2_O_2_-Induced Oxidative Stress and Blood–Milk Barrier Impairment in Dairy Cows

**DOI:** 10.3390/ijms24097742

**Published:** 2023-04-24

**Authors:** Chenchen Lyu, Bao Yuan, Yu Meng, Shuai Cong, Haoyu Che, Xingyu Ji, Haoqi Wang, Chengzhen Chen, Xinwei Li, Hao Jiang, Jiabao Zhang

**Affiliations:** 1Department of Laboratory Animals, Jilin Provincial Key Laboratory of Animal Model, Jilin University, Changchun 130062, China; cclv20@mails.jlu.edu.cn (C.L.); yuan_bao@jlu.edu.cn (B.Y.); mengy21@mails.jlu.edu.cn (Y.M.); congshuai19@mails.jlu.edu.cn (S.C.); chehy20@mails.jlu.edu.cn (H.C.); jixingyu0410@163.com (X.J.); hqwang19@mails.jlu.edu.cn (H.W.); chencz@jlu.edu.cn (C.C.); 2Key Laboratory of Zoonoses Research, College of Veterinary Medicine, Jilin University, Changchun 130062, China; lixinwei100@126.com

**Keywords:** puerarin, blood–milk barrier, BMECs, mastitis, oxidative stress

## Abstract

During the perinatal period, the bovine mammary epithelial cells of dairy cows exhibit vigorous metabolism and produce large amounts of reactive oxygen species (ROS). The resulting redox balance disruption leads to oxidative stress, one of the main causes of mastitis. Puerarin (PUE) is a natural flavonoid in the root of PUE that has attracted extensive attention as a potential antioxidant. This study first investigated whether PUE could reduce oxidative damage and mastitis induced by hydrogen peroxide (H_2_O_2_) in bovine mammary epithelial cells in vitro and elucidated the molecular mechanism. In vitro, BMECs (Bovine mammary epithelial cells) were divided into four treatment groups: Control group (no treatment), H_2_O_2_ group (H_2_O_2_ stimulation), PUE + H_2_O_2_ group (H_2_O_2_ stimulation before PUE rescue) and PUE group (positive control). The growth of BMECs in each group was observed, and oxidative stress-related indices were detected. Fluorescence quantitative PCR (qRT–PCR) was used to detect the expression of tightly linked genes, antioxidant genes, and inflammatory factors. The expression of p65 protein was detected by Western blot. In vivo, twenty cows with an average age of 5 years having given birth three times were divided into the normal dairy cow group, normal dairy cow group fed PUE, mastitis dairy cow group fed PUE, and mastitis dairy cow group fed PUE (*n* = 5). The contents of TNF-α, IL-6, and IL-1β in milk and serum were detected. In BMECs, the results showed that the PUE treatment increased the activities of glutathione (GSH), superoxide dismutase (SOD), catalase (CAT), and total antioxidant capacity (T-AOC); ROS and malondialdehyde (MDA) levels were reduced. Thus, PUE alleviated H_2_O_2_-induced oxidative stress in vitro. In addition, the PUE treatment eliminated the inhibition of H_2_O_2_ on the expression of oxidation genes and tight junction genes, and the enrichment degree of NRF-2, HO-1, xCT, and tight junctions (claudin4, occludin, ZO-1 and symplekin) increased. The PUE treatment also inhibited the expression of NF-κB-associated inflammatory factors (IL-6 and IL-8) and the chemokine CCL5 in H_2_O_2_-induced BMECs. In vivo experiments also confirmed that feeding PUE can reduce the expression of inflammatory factors in the milk and serum of lactating dairy cows. In conclusion, PUE can effectively reduce the oxidative stress of bovine mammary epithelial cells, enhance the tight junctions between cells, and play an anti-inflammatory role. This study provides a theoretical basis for PUE prevention and treatment of mastitis and oxidative stress. The use of PUE should be considered as a feed additive in future dairy farming.

## 1. Introduction

During the perinatal period, dairy cows undergo pregnancy, delivery, lactation, and other processes and are prone to oxidative stress due to high metabolic demand [1,2]. Specifically, the dynamics of hormones leading to the delivery and the beginning of lactation drastically change energy metabolism in the cows [3,4,5]. During the peripartum period, dairy cows face a dysfunctional immune system and an increased inflammatory state, resulting from a modulation of pathways related to metabolism, immune status, and endocrine system [6]. A growing body of evidence indicates that cows with ketosis or fatty liver display severe oxidative damage to the mammary glands [7,8]. Long-term oxidative stress can damage the mammary glands of dairy cows and impair the blood–milk barrier, resulting in declines in milk production and milk quality [9]. Furthermore, oxidative stress also weakens the resistance of breast tissue to foreign pathogens, thereby increasing the risk of mastitis in cows [10,11]. Thus, developing potential strategies to reduce oxidative stress may be an effective approach for preventing mammary gland injury and maintaining milk production in dairy cows.

During the perinatal period, a typical decline in dry matter intake (DMI) results in a negative energy balance (NEB) while maintaining a large amount of energy for fetal development and lactation [8]. Ketosis is often accompanied by oxidative stress, in part due to a significant increase in hepatic oxidative metabolism that produces free radicals such as reactive oxygen species [8,12,13]. When the production of ROS exceeds the antioxidant capacity of cells, oxidative stress occurs, leading to lipid peroxidation and protein modification disruption [14]. Bovine mammary epithelial cells act as an important line of defence against the invasion of mammary gland tissues by pathogenic microorganisms [15]. Oxidative stress can increase the expression of the inflammatory cytokines IL-6 and IL-8 and the inflammatory chemokine CCL5 in bovine mammary epithelial cells [16,17]. It impairs the tight junction proteins between bovine mammary epithelial cells and destroys the mammary barrier [16]. The breakdown of the blood–milk barrier and the continued accumulation of oxidative stress factors lead to a greater inflammatory response in the mammary gland, including redness, fever, abscess, and hardening of the mammary tissue [18,19]. Activation of phosphorylated p65, the nuclear factor κ-b (NF-κB) pathway, is a classical signaling pathway regulated in mastitis, and this upregulation may be caused by its overactivation.

Nuclear factor erythroid-2-related factor (Nrf2) is a nuclear transcription factor. Under normal conditions, Nrf2 remains sequestered in the cytoplasm via sulfhydryl-based interactions with Kelch-like ECH-associated protein 1 (Keap1) [20]. Oxidative stress occurs, and then the balance is broken. Nrf2 translocates into the nucleus and activates the transcription of a number of antioxidant genes, including glutathione, SOD and CAT, as well as other downstream antioxidant genes, such as HO-1 and cystine/glutamate antiporter (xCT) [21]. Nrf2 plays a protective role against oxidative stress induced by hydrogen peroxide (H_2_O_2_) in bovine mammary epithelial cells [22]. Pretreatment with baicalin can attenuate H_2_O_2_-induced oxidative stress in BMECs. *Moringa oleifera* leaf flavonoids can play a similar role by reducing intracellular catalase and NRF-2 concentrations [23]. *Moringa oleifera* leaf flavonoids can be used to treat flavonoids [24]. Tea polyphenols can be augmented with nuclear factor-erythrocyte 2-related factor 2 (NFE2L2) to alleviate ROS levels in BMECs [25], similar to lycopene. NFE2L2 also plays a role in reducing the oxidative stress level of cells [26].

In recent years, plant extracts have been widely used in feed additives to reduce the use of antibiotics due to their high efficiency, safety, and lack of drug residue [27]. Puerarin (PUE) is a flavonoid extracted from the plant Pueraria [28]. Recently, PUE has been shown to have a variety of pharmacological effects, including antioxidative, anti-inflammatory [29], antiviral [30], and antiapoptotic effects [31]. PUE can combat high glucose-induced damage to vascular endothelial cells by inhibiting the production of ROS and increasing the expression of tight junction genes [32]. Furthermore, PUE has been shown to activate the Nrf2 signaling pathway in mouse nerve cells [33], reducing oxidative damage and further inhibiting the activation of NF-κB in rat retinal pericytes [34]. However, the role of oxidative stress and inflammation in bovine mammary epithelial cells remains unknown. The purpose of this study was to investigate the effect of PUE on oxidative stress, inflammation, and tight junctions in bovine mammary epithelial cells and its potential molecular mechanism.

## 2. Results

### 2.1. Effect of PUE on the Viability of BMECs Treated with H_2_O_2_

CK18 staining was strongly positive, indicating that the BMECs had the characteristics of epithelial cells (Figure 1A). The cell viability was greater in the 10 μM, 20 μM, and 40 μM PUE groups but lower in the 80 μM PUE group than in the control group (Figure 1B; *p* < 0.05 and *p* < 0.01). The viability of BMECs decreased markedly in a dose-dependent manner in response to H_2_O_2_ (Figure 1C; *p* < 0.05 and *p* < 0.01) and was greater in the 20 μM and 40 μM PUE groups under H_2_O_2_ treatment than in the H_2_O_2_ group (Figure 1D; *p* < 0.05 and *p* < 0.01).

### 2.2. PUE Attenuates Oxidative Stress in H_2_O_2_-Treated BMECs

Compared with the control cells, BMECs treated with H_2_O_2_ for 24 h exhibited significantly higher levels of ROS (Figure 2A,B; *p* < 0.01) and MDA (Figure 2C; *p* < 0.01) but significantly lower values of T-AOC (Figure 2D; *p* < 0.01) and activity of SOD (Figure 2E; *p* < 0.01)), GSH (Figure 2F; *p* < 0.01)) and CAT (Figure 2G; *p* < 0.05). Compared with the H_2_O_2_ treatment group, the production of ROS and MDA was lower; nevertheless, the activities of T-AOC, SOD, GSH, and CAT were greater in the PUE and H_2_O_2_ treatment groups (Figure 2D–G; *p* < 0.05 and *p* < 0.01).

### 2.3. Effect of PUE on the Expression of Tight Junction Genes in H_2_O_2_-Treated BMECs

Compared with the control group, H_2_O_2_ treatment significantly reduced the mRNA abundance of *claudin-4* (Figure 3A; *p* < 0.01), *occludin* (Figure 3B; *p* < 0.01), and *ZO*-1 in BMECs after 24 h (Figure 3C; *p* < 0.01) as well as the mRNA abundance of *symplekin* (Figure 3D; *p* < 0.01). However, compared with H_2_O_2_ treatment, cotreatment with PUE and H_2_O_2_ resulted in higher mRNA levels of *claudin-4* (Figure 3A; *p* < 0.01), *occludin* (Figure 3B; *p* < 0.01), *ZO-1* (Figure 3C; *p* < 0.01), and *symplekin* (Figure 3D; *p* < 0.01).

### 2.4. Effects of PUE on the NF-κB Signaling Pathway in BMECs Treated with H_2_O_2_

Compared with the control, H_2_O_2_ treatment significantly increased the phosphorylation level of p65 in BMECs (Figure 4A,B; *p* < 0.01). PUE, however, significantly attenuated the H_2_O_2_-induced activation of p65 (Figure 4C; *p* < 0.01). Similarly, the expression of the NF-κB target genes *IL-6* and *IL-8* and the chemokine *CCL5* was increased following H_2_O_2_ treatment. However, the addition of PUE counteracted the induction of H_2_O_2_ in BMECs (Figure 4D–F; *p* < 0.01).

### 2.5. Effects of PUE on Nrf2 and Its Downstream Genes in BMECs Treated with H_2_O_2_

Furthermore, we examined the changes in genes associated with *Nrf2*. The results showed that treatment with H_2_O_2_ significantly reduced the mRNA abundance of *Nrf2* (Figure 5A; *p* < 0.05), *xCT*, and *HO-1* (Figure 4C and Figure 5B; *p* <0.05) in BMECs after 24 h compared to the control. However, the PUE treatment upregulated the mRNA levels of *Nrf2* and its downstream genes HO-1 and *xCT* in the H_2_O_2_-treated BMECs (Figure 5C; *p* < 0.01).

### 2.6. The Effect of PUE on the Prevention and Treatment of Mastitis in Dairy Cows

We added PUE to the dairy cow diet to explore its preventive and therapeutic effects on mastitis in dairy cows in vivo. As shown in the table, the levels of TNF-α, IL-1β, and IL-6 in whey from cows with mastitis were significantly higher than those from normal cows (Table 1; *p* < 0.01). When PUE was added to the dairy diet of the healthy cows and cows with mastitis, the levels of TNF-α, IL-1β, and IL-6 in the whey of the cows on day 7 were significantly lower than those on day 0 (Table 1; *p* < 0.01). We found that the expression of inflammatory factors in whey from the PUE-fed control healthy cows was lower than that from the mastitis cows (Table 1; *p* < 0.01) and decreased compared with the untreated controls. Similarly, serum levels of TNF-α, IL-1β, and IL-6 were significantly higher in mastitis cows than in the control cows (Table 2; *p* < 0.01). The serum TNF-α, IL-1β, and IL-6 levels of the cows with mastitis fed PUE root on day 7 were significantly lower than those on day 0 (Table 2; *p* < 0.01), but there was no change in the healthy cows after the 7th day of the PUE addition. Serum IL-6 levels in the PUE-fed control healthy cows were significantly lower than those in the mastitis group (Table 2; *p* < 0.01), and the expression of TNF-α and IL-1β also decreased, but there was no statistical significance. These results indicate that PUE added to the diet has an important preventive and therapeutic effect on mammary gland inflammation in dairy cows.

## 3. Discussion

During the lactation period, dairy cows have high metabolic rates, especially during the perinatal period and peak lactation [35]. The oxidative stress level was significantly increased in the mammary gland of dairy cows. In addition to the typical reductions in milk production, persistent oxidative stress affects lactation quality and reduces performance in dairy cows [7,36]. An increasing number of studies have shown that natural Chinese herbal extracts elicit protective effects against oxidative stress and inflammation, such as paeonol [37], Curcuminoids [38], and Houttuynia cordata Thunb [39]. In the present study, PUE alleviated oxidative injury and inflammation and enhanced tight junctions in bovine mammary epithelial cells. These data provide a theoretical basis for the prevention and treatment role of PUE on mammary oxidative injury in dairy cows.

Intracellular mitochondrial reactive oxygen species (ROS) can promote the development of oxidative stress [40] through changes in cell membrane integrity and function, which lead to damage to cells and organelles [41,42]. Studies have shown that flavonoids such as dihydromyricetin [43], dandelion flavonoids [44], and *Moringa oleifera* leaf flavonoids [24] can clear excessive intracellular ROS from dairy cow mammary glands and maintain intracellular redox balance [45]. In addition, PUE can protect human umbilical vein endothelial cells from apoptosis induced by oxidative stress [46]. These studies demonstrated that PUE has a beneficial effect on oxidative stress. However, whether PUE has an antioxidative effect in BMECs has not been verified. In this study, PUE attenuated H_2_O_2_-induced oxidative stress, as evidenced by reduced ROS and MDA levels in BMECs and further increased SOD, CAT and GSH activities. These data indicate that PUE has a protective effect against H_2_O_2_-induced oxidative stress in BMECs.

Production of large amounts of free radicals that cannot be eliminated causes inflammation. Oxidative stress and inflammation often have synergistic effects [47]. It was found that the occurrence of oxidative stress can activate various transcription factors, such as NF-κB, p53, PPAR-γ, and Nrf2 [48]. Activation of these transcription factors leads to the expression of many different genes, such as those encoding inflammatory cytokines, chemokines, and anti-inflammatory molecules [26]. Notably, lycopene activates the Nrf2-antioxidant response element (ARE) pathway and inhibits the NF-κB-related inflammation pathway. In this study, PUE inhibited the upregulation of the proinflammatory factors IL-6 and IL-8 and the chemokine CCL5 in H_2_O_2_-treated BMECs. The Western blot results showed that PUE reduced the phosphorylation of NF-κB induced by H_2_O_2_. In vivo, feeding PUE to cows with a high SCC score for one week reduced the serum levels of TNF-α, IL-1β, and IL-6, and the inflammatory factors in milk also showed the same downward trend, proving that PUE has preventive and therapeutic effects on inflammation in cows mammary. The results show that PUE can inhibit the activation of the NF-κB pathway and the expression of proinflammatory factors to ameliorate inflammation of the mammary glands.

During the development of the mammary glands, adjacent mammary epithelial cells are tightly connected to help establish and maintain milk synthesis and secretion [49]. The tight junctions of the mammary glands play an important role in maintaining the integrity of the blood–milk barrier [50]. In a mouse model of colitis, PUE inhibits intestinal epithelial dysfunction by increasing the expression of tight junction proteins [51]. Tight junctions are damaged during the degenerative stage of the breasts and when mastitis occurs. The mRNA levels of occludin, claudin, and zo-1 are downregulated under high-temperature exposure [52] and simultaneous LPS and high-temperature stimulation in bovine mammary epithelial cells [53]. In this study, the mRNA levels of four tight junction genes, occludin, claudin, ZO-1, and symplekin, were upregulated after the PUE treatment in an oxidative stress cell model. Therefore, PUE can improve the tight junctions of breast epithelial cells and partially protect the integrity of the breast barrier.

Nrf2 can promote the expression of downstream antioxidant proteins and enhance the antioxidant capacity of cells. xCT is a cystine transporter that regulates the synthesis of GSH [54]. HO-1 is an important antioxidant enzyme [55]. Notably, both are regulated by Nrf2 [56]. In our research, the PUE treatment upregulated the mRNA levels of Nrf2 and its downstream antioxidant enzymes HO-1 and xCT in BMECs. Meanwhile, the increase in TNF-α, IL-1β, and IL-6 and the decrease in occludin, claudin, ZO-1, and symplekin induced by H_2_O_2_ in mammary epithelial cells were recovered. Then, the high SCC score of cows was reduced, the blood–milk barrier is repaired to relieve mammary gland inflammation in dairy cows. In summary, in vitro and in vivo, the PUE treatment alleviated oxidative stress caused by hydrogen peroxide by upregulating NRF2; at the same time, inhibition of NF-κB pathway activation and weakening of tight junction damage were observed (Figure 6).

## 4. Materials and Methods

### 4.1. Cell Culture and Processing

Cow mammary tissue blocks were obtained by puncturing the animals, and they were then cleaned using PBS + penicillin − streptomycin (HyClone, South Logan, UT) 3–4 times. Blocks of 1 mm^3^ mammary tissue were sliced into pieces and put into cell culture flasks. The cell culture flasks were inverted in a cell culture incubator for around 6 h. We waited for BMECs to climb out from the tissue mass before adding DMEM/F12 (HyClone, UT, USA) containing 10% fetal bovine serum (Gibco, Grand Island, NY, USA) to the cell culture flask. Trypsin digestion was employed for purification two to three times after BMECs have a large crawled out. Cytokeratin 18 (Abcam, Cambridge, UK) was used to identification mammary epithelial cells so that they could be recognized. Mammary epithelial cells that had been identified were grown, passaged, and frozen. The BMECs employed in this experiment ranged in generation from third to sixth.

BMECs were inoculated in 25 cm culture flasks and cultured in a 5% CO_2_ incubator at 37 °C. The medium was DMEM/F12 containing a 1% penicillin-streptomycin mixture and 10% fetal bovine serum. Cells were assayed when their growth density reached the 80% area, and they were passaged or subjected to further treatment. A stock solution of 5 mg/mL hydrogen peroxide (Kaiyuan Chemical Reagent Plant 1) was mixed with the medium and prepared for immediate use. In addition, a stock solution of 4 mg/mL PUE (Solarbio, Beijing, China) was dissolved in dimethyl sulfoxide (DMSO) and stored in the dark at −20 °C. All experimental treatments were carried out using starvation medium.

### 4.2. Animals and Design

This study was conducted on a commercial dairy herd in Qiqihar, China. Lactating dairy cows were housed in individual tie stalls. Lactating cows were kept in separate tethering pens, aged about 5 years, and had a mean body weight (BW) of 624 ± 63 kg with parity = 3. All cows had no history of long-term disease and all fetuses delivered were normal. Screenings for mastitis-positive dairy cows and healthy dairy cows were performed by the California mastitis test (CMT) [57]. Ten normal dairy cows and ten dairy cows with clinical mastitis were used to study the effect of PUE on dairy mastitis by feeding or not feeding it. The cows were randomly divided into the normal cow group (Control), the normal cows fed PUE group (Control + PUE), the mastitis cow group (Mastitis), and the mastitis cows fed PUE group (Mastitis + PUE). There were five cows in each group. PUE was added by feeding 300 g PUE mixed with a standard diet every day for 7 consecutive days, and the animals were housed individually to ensure comparable feed intake. The cows were provided with unlimited access to fresh water and given free range following feeding. All cows received no additional medication throughout the experiment. Changes in daily somatic cell number (SCC), milk yield, and milk composition (Fat, Protein, Lactose, and other solids) were recorded for all the selected cows from day 2 to 7 of the experiment. These results are presented in Appendix A to ensure the accuracy of treatment assignments.

On days 0 and 7 of the feeding experiment, 50 mL of milk samples and 10 mL of blood were collected. Milk from different groups of cows was collected from the same mammary gland. Before milking, hands were washed and disinfected before contacting the mammary gland of the cow. After disinfecting the mammary of the cow, milk was collected. The first three times of milk were discarded during each milk collection to maintain aseptic operation and complete labelling of the collected milk. The collected milk was stored at 4 °C for subsequent experimental operations, such as whey separation. A similar procedure was used for blood collection. That is, hands were kept clean and sterile before blood collection. After the oxtail vein was wiped and disinfected with iodophor and alcohol cotton, blood was collected using a collection vessel (Hongda, Jiangxi, China) without anticoagulant agent. After blood collection, the vessel was marked, and the blood sample was stored at 4 °C for subsequent test operations, and then the serum was collected using centrifugation at 2000 rpm for 2 min. All collected milk and blood must be marked with the corresponding cow number, milk area, and collection date.

### 4.3. Immunofluorescence

BMECs were uniformly inoculated in 6-well plates for 1 d. Immunofluorescence staining of cells was performed according to standard procedures. Briefly, phosphate buffer (PBS) was washed three times, and 4% cold paraformaldehyde was fixed for 20 min followed by three washes of PBS. A 0.2% Triton X-100 solution was used as the cell permeabilization solution for a total of 10 min. After washing the cells with PBS, goat serum was added dropwise, and the cells were incubated at room temperature for 30 min. The cells were incubated with a primary antibody targeting CK18 (1:500, ab137860, Abcam, Cambridge, UK) at 4 °C overnight, followed by three washes with PBST for 5 min each. Anti-rabbit IgG (1:1000, 4412; CST, Boston, MA, USA) was used for 1 h. Finally, photos were collected after staining with Hoechst 33,342 (Kegan Biotech, Nanjing, China) for 5 min and washing.

### 4.4. Cell Viability Determination

BMECs were inoculated in 96-well plates at a density of 2 × 10^4^ cells/well, and 100 μL of complete medium was added for the cell viability assay. Briefly, the cell density was changed to a serum-free medium after reaching 70~80%. Cells were treated with different concentrations of H_2_O_2_ (0, 200, 400, 600, 800, 1000 μM) for 24 h. Then, 10 μL CCK-8 solution was added and incubated at a constant temperature of 37 °C for 2 h. In addition, cells were treated with different concentrations of PUE (0, 10, 20, 40, 80, 160 μM) for 24 h in agreement with the above. The final results of cell viability were calculated by detecting the absorbance of cells at 450 nm using an enzyme marker (Tecan, Safire, Austria). In subsequent experiments, BMECs were treated with PUE at a concentration of 40 μM and H_2_O_2_ at a concentration of 600 μM for 24 h.

### 4.5. ROS Determination

ROS levels in BMECs were detected using the fluorescent probe DCFH-DA (S0033 M, Beyotime, Shanghai, China). Cells were co-incubated with 10 μmol/L DCFH-DA for 20 min at 37 °C under serum-free culture conditions protected from light. Then, the cells were washed three times to remove the probes that did not successfully enter the cells. Photographs were collected under a fluorescence microscope (Nikon, Japan).

### 4.6. Detection of Oxidative Stress and Antioxidative Stress Indicators

BMECs were inoculated in 6-well plates and replaced with a serum-free medium when the fusion level reached approximately 70%. H_2_O_2_ and PUE were added for coincubation for 24 h. We used a T-AOC assay kit (BC1315), SOD assay kit (BC0175), H_2_O_2_ assay kit (BC0205), GSH assay kit (BC1175), and malondialdehyde (MDA) detection kit (BC0025) to measure T-AOC, SOD, CAT, GSH activity, and MDA levels. All kits were provided by Beijing Solarbio Science & Technology Co., Ltd., Beijing, China. The assays were performed according to the manufacturers’ instructions.

### 4.7. RNA Extraction and Real-Time Fluorescent Quantitative PCR (qRT–PCR)

Total RNA was extracted from cells using TRIzol (Invitrogen, Carlsbad, CA, USA), and the RNA concentration was measured using a Nanodrop 2000 spectrophotometer (Thermo, Waltham, MA, USA). Reverse transcription reagents were purchased from Tiangen Science & Technology Co., Ltd. (Beijing, China), cDNA synthesis was performed according to the instructions, and qRT–PCR was performed according to the instructions of SuperReal PreMix Plus (SYBR Green; Tiangen) using a CFX96 Real-Time PCR system (Bio-Rad, Hercules, CA, USA). The following table lists the primers used in qRT–PCR. Finally, the experimental data were analyzed with the relative quantitative method (2^−ΔΔCT^), with *β-actin* as the internal reference gene. The primers used in this article were provided by Sangon Biotech (Shanghai) Co., Ltd., Shanghai, China. All the primers applied are listed in Appendix A.

### 4.8. Western Blot Analysis

Total protein was isolated using a radio immunoprecipitation assay (RIPA) lysis buffer (Solarbio, Beijing, China), quantified using a BCA kit (P0012S, Beyotime, Shanghai, China), and separated via 10% SDS–PAGE. Then, the proteins were transferred to a polyvinylidene fluoride (PVDF) membrane (Millipore, Bedford, MA). PVDF membranes were blocked for 1 h with TBST in dissolved 5% bovine serum albumin (BSA, Sigma, Milpitas, CA, USA). Primary antibodies (p65 and P-p65, Cell Signaling Technology, USA, 1:1000; GAPDH, Bioworld, 1:300) were used overnight at 4 °C. PVDF membranes were washed with TBST and incubated with IgG antibodies in common for 1 h. Chemiluminescent substrates (Tanon, Shanghai, China) were applied for protein band visualization, and ImageJ software was used for quantification.

### 4.9. Enzyme-Linked Immunosorbent Assay

Bovine TNF-α (SEKB-0303-96T), Bovine IL-6 (SEKB-0365-96T), and Bovine IL-1β (SEKB-0363-96T) kits were purchased from Solarbio, China. The intra- and interassay variability were <15% for the ELISA kits. All reagents and standards were prepared as directed. The plate was washed three times before the assay. Then, 100 µL standard or samples was added to each well and shaken with a micro-oscillator (100 r/min) for 60 min at room temperature, and it was aspirated and washed four times. A 100 µL working solution of biotin-conjugated anti-bovine IL-1β antibody was added to each well, shaken with a micro-oscillator (100 r/min), and incubated for 1 h at room temperature, and it was aspirated and washed four times. Then, 100 µL of a working solution of streptavidin-HRP was added to each well, which was shaken with a micro-oscillator (100 r/min) and incubated for 20 min at room temperature. Then, 100 µL of substrate solution was added to each well, the plate was incubated for 10 min at room temperature, and it was protected from light. Then, 50 µL of stop solution was added to each well. Finally, a microplate reader (Tecan, Safire, Austria) was used to detect the absorbance of the cells at 450 nm within 5 min. The TNF-α and IL-6 inspection methods were the same as those for IL-1β.

### 4.10. Statistical Analysis

All experimental results were repeated three times, and experimental data are expressed as the mean ± SD deviation. SPSS 22.0 software (SPSS Inc., Chicago, IL, USA) performed a normality test (*p* > 0.5), then comparisons were able to be made between the two groups and were performed by unpaired two-tailed Student’s t test. * *p* < 0.05 was considered to indicate statistical significance, ** *p* < 0.01 was considered to indicate extreme significance, and NS *p* > 0.05 was considered to indicate a lack of significance.

## 5. Conclusions

The results, obtained under our experimental conditions, show that PUE therapy can reduce the production of ROS and MDA, increase the expression of Nrf2 and the activity of antioxidant enzymes, and protect the integrity of the blood–milk barrier to effectively inhibit oxidative stress and inflammation in H_2_O_2_-treated BMECs. In vivo experiments have shown that feeding PUE to cows with a high SCC score reduces the levels of inflammatory cytokines in blood and milk. PUE can be considered a feed additive in future dairy farming.

## Figures and Tables

**Figure 1 ijms-24-07742-f001:**
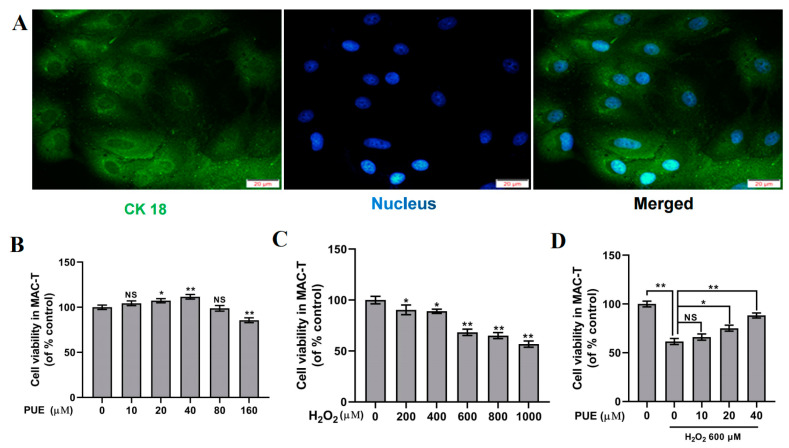
Effect of PUE on the viability of BMECs treated with H_2_O_2_. (**A**) Identification of CK18 in BMECs by immunofluorescence staining. Nuclei, blue; CK18, green; Scale bar = 20 μm. (**B**) Effect of H_2_O_2_ on cell viability under normoxic conditions. (**C**) Effect of PUE on cell viability under normoxic conditions. (**D**) Effect of PUE within the safe concentration range on the viability of BMECs treated with H_2_O_2_. All experiments were performed in triplicate. The data are shown as the means ± SDs. * *p* < 0.05, ** *p* < 0.01, NS *p* > 0.05 (not significant), unpaired Student’s *t* test.

**Figure 2 ijms-24-07742-f002:**
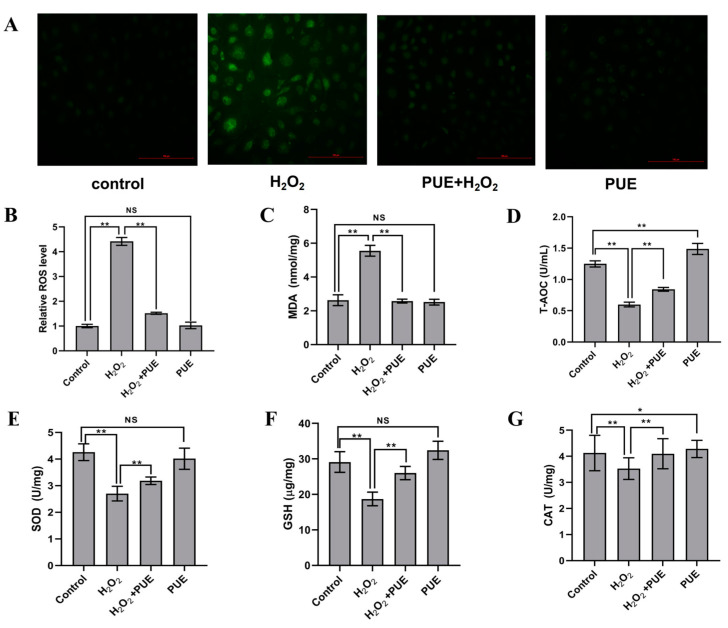
PUE prevents H_2_O_2_-induced oxidative stress in BMECs. (**A**,**B**) BMECs were treated with PUE (40 μM) and/or H_2_O_2_ (600 μM) for 24 h. ROS levels were detected by fluorescence microscopy; Scale bar = 100 μm. (**C**) MDA content in BMECs. (**D**) T-AOC in BMECs. (**E**) SOD activity in BMECs. (**F**) GSH activity in BMECs. (**G**) CAT activity in BMECs. All experiments were performed in triplicate. The data are shown as the means ± SDs. * *p* < 0.05, ** *p* < 0.01, NS *p >* 0.05 (not significant), unpaired Student’s *t* test.

**Figure 3 ijms-24-07742-f003:**
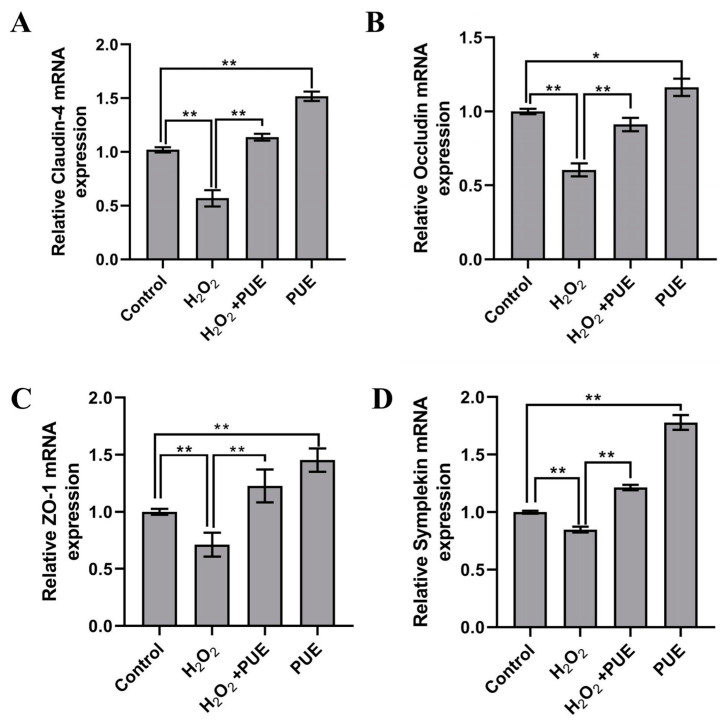
Effect of PUE on the mRNA levels of tight junction genes in H_2_O_2_-treated BMECs. The mRNA levels of tight junction genes were measured by qRT–PCR. (**A**) *Claudin-4* mRNA level. (**B**) *Occludin* mRNA level. (**C**) *ZO-1* mRNA level. (**D**) *Symplekin* mRNA level. All experiments were performed in triplicate. The data are shown as the means ± SDs. * *p* < 0.05, ** *p* < 0.01, unpaired Student’s *t* test.

**Figure 4 ijms-24-07742-f004:**
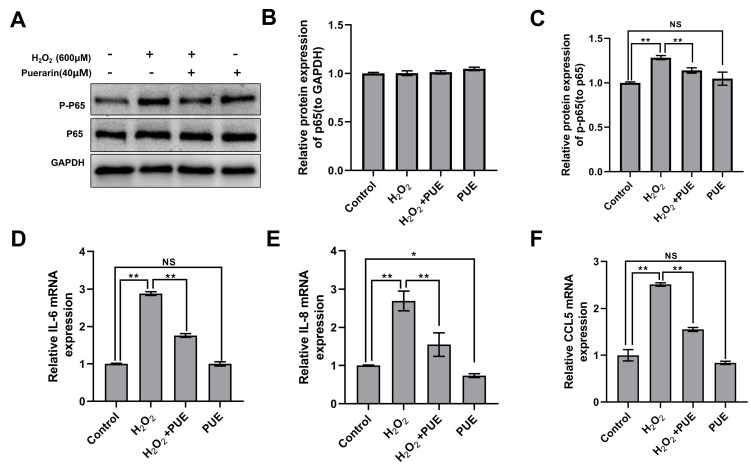
PUE in medium weakens H_2_O_2_−induced increases in P-p65 in BMECs. Cells were collected after incubation with H_2_O_2_ and/or PUE for 24 h. (**A**) Detection of P-p65 and p65 levels by Western blotting. GAPDH was used as the loading control. (**B**,**C**) P-p65 and p65 levels were normalized to p65 and GAPDH levels, respectively. (**D**) *IL-6* mRNA levels in BMECs. (**E**) *IL-8* mRNA levels in BMECs. (**F**) *CCL5* mRNA levels in BMECs. All experiments were performed in triplicate. The data are shown as the means ± SDs. * *p* < 0.05, ** *p* < 0.01, NS *p >* 0.05 (not significant), unpaired Student’s *t* test.

**Figure 5 ijms-24-07742-f005:**
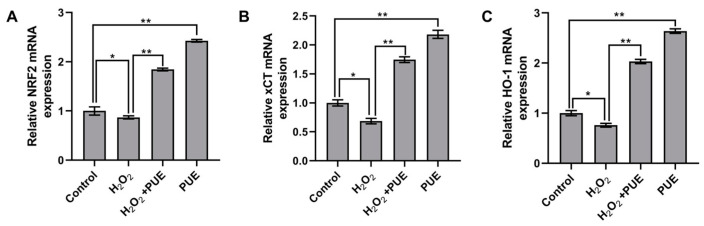
Effect of PUE on *Nrf2* and its downstream genes in H_2_O_2_-treated BMECs. BMECs were treated with (600 μM) H_2_O_2_ in the presence or absence of 40 μM PUE for 24 h, and the expression levels of Nrf2. (**A**) and its downstream genes *xCT*, (**B**) and *HO-1*, (**C**) were measured by qRT–PCR. All experiments were performed in triplicate. The data are shown as the means ± SDs. * *p* < 0.05, ** *p* < 0.01, unpaired Student’s *t* test.

**Figure 6 ijms-24-07742-f006:**
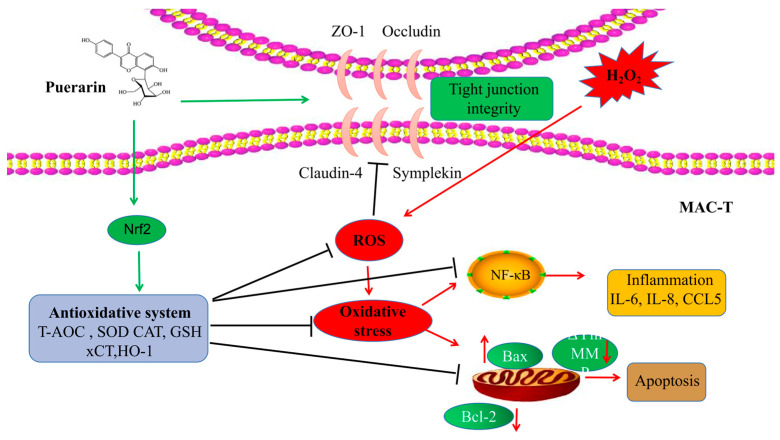
Schematic diagram of the proposed model. PUE reduces the generation of ROS, maintains the intracellular redox balance, protects the blood–milk barrier, and reduces H_2_O_2_-induced inflammation and oxidative stress in breast epithelial cells.

**Table 1 ijms-24-07742-t001:** Expression of inflammatory factors after PUE supplementation for prevention and treatment in milk. (*n* = 5).

	Time	Control ^1^	Control + PUE ^2^	Mastitis ^3^	Mastitis + PUE ^4^
IL-6	0 d	130.58 ± 42.99	146.54 ± 36.02	513.82 ± 32.66 ^b^	492.21 ± 43.31 ^b^
	7 d	120.29 ± 34.55	116.23 ± 33.79 ^a^	489.86 ± 39.86 ^b^	303.18 ± 27.93 ^abc^
IL-1β	0 d	10.54 ± 0.89	10.21 ± 1.38	37.88 ± 2.25 ^b^	36.67 ± 0.75 ^b^
	7 d	10.58 ± 0.90	9.87 ± 0.78 ^a^	35.71 ± 2.14 ^b^	21.42 ± 0.85 ^abc^
TNFα	0 d	97.15 ± 8.83	103.15 ± 2.51	175.61 ± 7.95 ^b^	163.75 ± 6.01 ^b^
	7 d	98.20 ± 1.14	98.20 ± 6.74 ^a^	182.37 ± 20.24 ^b^	128.35 ± 4.51 ^abc^

^1^ Healthy cows without any treatment. ^2^ Healthy cows fed a diet supplemented with PUE. ^3^ Diseased cows with mammary gland inflammation. ^4^ Mastitis cows with a diet supplemented with PUE. ^a^ indicates a significant difference in the corresponding protein expression on day 0 and day 7 in the same group (*p* < 0.01), ^b^ indicates a significant difference relative to the control group at the same time (*p* < 0.01), and ^c^ indicates a significant difference with the mastitis group at the same time (*p* < 0.01). All experiments were repeated three times, and experiments were performed simultaneously using different individual cows. The data are shown as the mean ± SD. Allowed comparisons between the two groups were performed by unpaired two-tailed Student’s *t* test.

**Table 2 ijms-24-07742-t002:** Expression of inflammatory factors in cow serum after PUE supplementation for prevention and treatment. (*n* = 5).

	Time	Control ^1^	Control + PUE ^2^	Mastitis ^3^	Mastitis + PUE ^4^
IL-6	0 d	124.62 ± 8.76	142.74 ± 11.42	530.47 ± 24.15 ^b^	516.74 ± 44.23 ^b^
	7 d	134.12 ± 2.55	115.66 ± 9.94 ^a^	522.67 ± 26.65 ^b^	332.03 ± 19.09 ^abc^
IL-1β	0 d	14.54 ± 4.33	15.08 ± 3.00	49.83 ± 3.58 ^b^	50.92 ± 2.58 ^b^
	7 d	17.08 ± 0.90	13.92 ± 3.19	53.29 ± 2.06 ^b^	30.10 ± 3.12 ^abc^
TNFα	0 d	149.63 ± 10.89	157.00 ± 3.08	239.60 ± 7.23 ^b^	234.49 ± 7.45 ^b^
	7 d	146.82 ± 7.50	142.72 ± 15.16	240.50 ± 0.87 ^b^	186.79 ± 11.87 ^abc^

^1^ Healthy cows without any treatment. ^2^ Healthy cows fed a diet supplemented with PUE. ^3^ Diseased cows with mammary gland inflammation. ^4^ Mastitis cows with a diet supplemented with PUE. ^a^ indicates a significant difference in the corresponding protein expression on day 0 and day 7 in the same group (*p* < 0.01), ^b^ indicates a significant difference relative to the control group at the same time (*p* < 0.01), and ^c^ indicates a significant difference with the mastitis group at the same time (*p* < 0.01). All experiments were repeated three times, and experiments were performed simultaneously using different individual cows. The data are shown as the mean ± SD. Allowed comparisons between the two groups were performed by unpaired two-tailed Student’s *t* test.

## Data Availability

Not applicable.

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
