# Peer review of "Puerarin Alleviates H2O2-Induced Oxidative Stress and Blood–Milk Barrier Impairment in Dairy Cows"

_ijms, 2023, doi:10.3390/ijms24097742_

Round 1

Reviewer 1 Report (Previous Reviewer 4)

The authors did not bother to supplement the data for the in vivo study, nor to explain the origin of the samples for the in vitro study. There is still insufficient data on cows, nutrition and characteristics of mastitis (clinical, subclinical and etiological diagnosis). There is no additional evidence that there was mastitis other than the SCC, which is an indicative test, and an increase in the SCC does not mean the presence of mastitis.

Are the samples for the in vitro study derived from animals from the in vivo study? This is another major inconsistency of this experiment, which needs to be explained.

For the purposes of the in vivo study, it is essential to present data on milk quality and clinical findings or to remove the word mastitis from the name, but to talk about cows with a high SCC score.

The conclusion is too ambitious, since we have no data on the diagnosis of mastitis or data on the cure of cows, we cannot talk about the therapeutic effect of PUE in the treatment of mastitis. We can talk about anti-inflammatory and antioxidant effects. In order to talk about the therapeutic effect, it is necessary to carry out precise dosing, prove PUE residues in milk and udder tissue, etc. Therefore, the conclusion about the therapeutic effect is not correct, because this is not a clinical study by any of its characteristics.

Author Response

On behalf of all the contributing authors, I would like to express our sincere appreciation of your letter and reviewers’ constructive comments concerning our article entitled “Puerarin alleviates H2O2-induced oxidative stress and blood-milk barrier impairment in dairy cow” (Manuscript ijms-2362357). These comments are all valuable and helpful for improving our article.

According to the associate editor and reviewers’ comments, we have made extensive modifications to our manuscript and supplemented extra data to make our results convincing. In this revised version, changes to our manuscript were all highlighted within the document by using yellow text. Point-by-point responses to the nice associate editor and two nice reviewers are listed below this letter. We appreciate for Reviewers’ warm work earnestly and hope that the correction will meet with approval.

Reviewer 2 Report (Previous Reviewer 5)

Manuscript ijms-2362357

The paper entitled "Puerarin alleviates H2O2-induced oxidative stress and blood milk barrier impairment in dairy cows" is an experimental study that aims to test the effect of Puerarine PUE on oxidative stress, inflammation, and tight junctions in bovine mammary epithelial cells and its potential molecular mechanism.

The study, also includes a field trial, conducted on four groups of dairy cows, either healthy or affected by mastitis, to which PUE was added to the daily feed ration. The effects of PUE, obtained under these experimental conditions, are potentially interesting, but should be replicated on a larger scale.

Below are my minor comments, line by line

line 294: "and gave birth to three litters". I suggest replacing it "with parity = 3."

line 306-307: "mammary gland area". The term "area" appears redundant and inappropriate, also check elsewhere in the text.

line  317: "..... for subsequent test operations. And ...". Rewrit it as "..... for subsequent test operations, and ...".

line 405: "The results demonstrate that PUE therapy can reduce the production of ROS and MDA, ........". I suggest to rewrite it as "The results, obtained under our experimental conditions, show that PUE therapy ...."

Minor editing of English language required.

Author Response

On behalf of all the contributing authors, I would like to express our sincere appreciation of your letter and reviewers’ constructive comments concerning our article entitled “Puerarin alleviates H2O2-induced oxidative stress and blood-milk barrier impairment in dairy cow” (Manuscript ijms-2362357). These comments are all valuable and helpful for improving our article.

According to the associate editor and reviewers’ comments, we have made extensive modifications to our manuscript and supplemented extra data to make our results convincing. In this revised version, changes to our manuscript were all highlighted within the document by using yellow text. Point-by-point responses to the nice associate editor and two nice reviewers are listed below this letter. We appreciate for Reviewers’ warm work earnestly and hope that the correction will meet with approval.

Round 2

Reviewer 1 Report (Previous Reviewer 4)

I am looking forward to your future research in which you will diagnose mastitis much more precisely and in which you will perform a real pharmacological study with PUE with indicators of pharmacokinetics and pharmacodynamics. I strongly encourage you in this!

This manuscript is a resubmission of an earlier submission. The following is a list of the peer review reports and author responses from that submission.

Round 1

Reviewer 1 Report

This manuscript described the effects of puearin treatment on bovine mammary epithelial cells in vitro in cell culture. The study design is fine, the methods are well described and results are well presented.

There are only minor point that should be clarified:

In the Abstract:

- the abbreviation BMEC should be explain when firstly mentioned.

- the investigated groups should be better described (line 19-20)

Reviewer 2 Report

General comments

In the study entitled “Puerarin alleviates H2O2-induced oxidative stress and bloodmilk barrier impairment in cow mammary gland” the Authors investigated whether puerarin could reduce oxidative damage and mastitis in duced by hydrogen peroxide (H2O2) in bovine mammary epithelial cells in vitro and elucidated the molecular mechanism.

 This study provides a theoretical basis for Puerarin prevention and treatment of mastitis and oxidative stress. According to the findings obtained in the study, Authors concluded that PUE can effectively reduce the oxidative stress of bovine mammary epithelial cells, enhance the tight junctions between cells, and play an anti-inflammatory role.

This study improve the knowledge on the field. I think that the subject of the work is of interest and that the topic of the manuscript is appropriate for the Journal. The information is of significant interest to the Journal's readers. I suggest some changes in order to improve the text.

Specific comments

I think that the title well reflects the main aim and findings of the work.

The abstract adequately summarize results and significance of the study, and, the keywords represent the article adequately. However, Authors should add some information on animals enrolled in the study (i.e. the number of enrolled animals, age, body weight).

The introduction section is well written and it falls within the topic of the study, and Authors cited appropriately bibliographic information. I think that Authors should better emphasize the features of peripartum period with focus on adaptation of animals during this particular life phase. Indeed, it is well known that during peripartum period dairy cows are generally under a state of negative energy balance which lead to weakened immune system and, thus, it make cows more susceptible to infections. In view of this, please enhance the sentence (Lines 42- 43) “During the perinatal period, dairy cows have undergone pregnancy, delivery, lactation and other processes and are prone to oxidative stress due to high metabolic demand [1, 2].” by adding insights on inflammation during peripartum period in dairy cows.

On this regards, I suggest to modify the sentence as following “During the perinatal period, dairy cows have undergone pregnancy, delivery, lactation and other processes and are prone to oxidative stress due to high metabolic demand [1, 2]. Specifically, the dynamics of hormones leading to the delivery and the beginning of lactation drastically change energy metabolism in the cows (Piccione G. et al., Journal of Dairy Research, 2011, 78: 421-425; Fiore E. et al., Archiv Tierzucht 57, 3: 1-9, 2014; Fiore E. et al., Anim. Prod. Sci.  57 (6), 1007–1013). During peripartum period, dairy cows face a dysfunctional immune system and an increased inflammatory state. A modulation of pathways related to metabolism, immune status and endocrine system (Arfuso F. et al., Theriogenology 196, 2023, 157-166).”

The section of Materials and Methods is clear for the reader and it meticulously describes the methods applied in the study. However, some information should be added.

In the subsection “4.8. Animals and design” Authors should add more information on animals: What about the calving? Did cows deliver healthy, viable full-term calves, without assistance?

Regarding blood collection, Authors should indicate the tubes used for blood collection (type (without anticoagulant agent), Manufacture and country). Authors should indicate whether blood sampling was performed at the same hour of the day from each animal. Also, centrifugation features applied to obtain serum samples should be indicated.

In the subsection “4.9. Enzyme-linked immunosorbent assay” Authors should indicate whether the kits were specific for bovine species, or whether they were previously validated for the studied species. Moreover,  for ELISA analyses, Authors should indicate the intra- and inter-assay variability for each Interleukin.

Regarding statistical analysis, Did Authors perform a normality test in order to test the normal distribution of data? Please clarify this aspect.

Results section as well as Discussion section is clear and well written, the findings obtained in the study were well discussed and justified with appropriate references.

In the conclusion section Authors well summarize the main results of the study and emphasize the significance, however, I suggest to delete “In summary” at the beginning of the section and to avoid the use of personal form (i.e. our, we etc.).

Tables are generally good and Figures are nice representing well the results gathered in the study.

Reviewer 3 Report

the corrections in file.

Reviewer 4 Report

Modern pharmacological studies have shown that puerarin has a variety of bioactive effects, for instance, estrogen-like activity, anti-inflammation, antioxidant response, blood pressure control, blood glucose reduction, and cancer reduction. In humans, puerarin has also been recommended in several clinical trials for the treatment of inflammatory diseases, including acute tonsillitis, chronic bronchitis, and ulcerative colitis.

How did the cows ingest PUE? Are you sure that the necessary and sufficient quantities have been entered? Do you think the inflammatory and antioxidant response would depend on the dose of PUE and why the trial was not done with different doses of PUE. Different doses acting in different ways would guarantee that the changes were solely due to PUE. Another important point is the cumulative effect of PUE, so it is not clear whether the effects would be better if the treatment with PUE was continued for 7 or 15 days. Has there been a cure over time, self-healing, or is PUE the new cure for mastitis?

Because of all the above, it is necessary to determine the results of the CMT test for all days of the test, and it would be good to get acquainted with the number of somatic cells, as well as the basic composition of milk (protein, fat, lactose, possibly fatty acids).

A missing set of data is which causative agents of mastitis have been proven. There are different pathogenic mechanisms in different causative agents of mastitis. Some mastitis are self-healing. It is necessary to determine the age of the cows, in what period of lactation they are and how much milk they gave in the previous lactation. These characteristics of cows may lead to a confounding effect on the variables, as it is known that in early lactation there is oxidative stress and immunosuppression in cows.

Has a certain therapy been performed for mastitis in cows? Did the cows receive other types of antioxidants and what was the composition of the ration?

In MandM, it is first necessary to describe the animals, the management of the experiment, the design, the feeding and care conditions, the conditions on the farm, etc. Then we should move on to sampling and then to the methods that were applied.

Was the milk sampled from the affected udder quarters or from all quarters. How was it handled if several quarters were affected?

Reviewer 5 Report

Manuscript ID ijms-2285436

I had the opportunity to review the paper, entitled “Puerarin alleviates H2O2 induced oxidative stress and blood milk barrier impairment in cow mammary gland”.

In this study, the authors investigate whether puerarin PUE (natural flavonoid extracted from the root of Pueraria mirifica L.) could reduce oxidative damage and mastitis induced by H2O2 in bovine mammary epithelial cells (BMECs) in vitro and elucidated the molecular mechanism.

In vitro experiment: BMECs were divided into four treatment groups: a) control group, b) stimulation group (H2O2), c) rescue group (PUE + H2O2) and d) positive control group (PUE). The growth of BMECs in each group was observed, and oxidative stress-related indices were detected.

PCR (qRT-PCR) was used to detect the expression of closely related genes, antioxidant genes and inflammatory factors while Western Blot was used for evaluate the expression of p65 protein.

In vivo experiment: 20 dairy cows were divided in 4 groups, 1) normal dairy cow group, 2) normal dairy cow group fed PUE, 3) mastitis dairy cow group fed PUE, and 4) mastitis dairy cow group fed PUE. TNF-α, IL-6 and IL-1β contents were detected in milk and blood serum.

The authors found that PUE alleviated H2O2 induced oxidative stress in vitro, in addition, PUE treatment eliminated the inhibition of H2O2 on the expression of oxidation genes and tight junction genes.

Authors to conclude that PUE can effectively reduce the oxidative stress of BMECs, enhance the tight junctions between cells, and play an anti-inflammatory role and suggest to dairy farmers using PUE as a feed additive in future.

The paper covers topics included in the aim of the Journal and is interesting, but I have to make two separate judgments for the experiments conducted "in vitro" and "in vivo" respectively.

The "in vitro" experiment was well conducted by the authors, obtaining very interesting results, while the "in vivo" experiment on dairy cows was conducted incorrectly, for the reasons that I summarize below:

1) Mastitis is an inflammation of the mammary gland, induced by bacteria, algae, fungi etc.

2) Mastitis can be diagnosed (almost always) from milk, taken aseptically, from all 4 teats of the mammary gland. Milk samples should be analyzed possibly following the specifications of the National Mastitis Council (Direct method); see NMC Microbiological Procedures for the Diagnosis of Bovine Udder Infection (3rd ed.), National Mastitis Council, Arlington, VA (1999).

3) Mastitis induces changes in the qualitative and cytological (somatic cells) milk composition.

4) Somatic cells are thus an "indirect" parameter for the evaluation of mastitis, although their content may increase due to causes other than pathogen infections (teat trauma, oestrus phase, etc.).

5) Determination of "somatic cells number" contained in milk is done with automated instruments, while the California Mastitis Test (CMT) is a common reagent (useful in routine barn practice) that classifies milk using a 5-point scale (negative, trace, 1 positive, 2 positive, 3 positive), to which increasing values of somatic cells correspond.

Therefore, the criteria and methods used for dividing dairy cows into 4 groups, particularly the "mastitis" group, with CMT criteria is incorrect. The diagnosis of “mastitis” was to be made by the “direct method” by isolating the microorganism causing mastitis (see NMC 1999). In addition, milk should be sampled from the 4 teats of the udder and analyzed daily during the test (day 0, 1, .... 7) for each animal.

Below I list my comments line by line

Line 53: "During the perinatal period, a typical decline in DMI results in a NEB while"; specify the meaning of "DMI" and "NEB".

Line 76: may attenuate H2O2 induced oxidative stress in BMECs. Moringa oleifera leaf flavonoids; Rewrite H2O2 in the correct form; write Moringa oleifera in italics (ceck elsewhere in the text).

Line 84: Puerarin (PUE) is a flavonoid extracted from the natural plant Pueraria. Check the term "natural".

Line 92: The aim of this study was to investigate the effect of PUE on oxidative stress, inflammation and tight junctions in bovine mammary epithelial cells and its potential molecular mechanism. An 'in vivo' study on four groups of dairy cows is not indicated.

Line 106-111: I suggest representing the differences between the mean values (p<0.05) between the histograms with letters instead of asterisks. applying the same criterion to all graphs in the text.

Line 184: Table 1, the letters indicating significance are not entered correctly. how is it possible to find between 2 mean values three letters? (a vs abc). Recheck all tables.

Line 275: CO2 incubator; check.

Line 282: DMSO and stored in the dark at -20°C. explain the meaning of DMSO.

Line 343: "4.8. Animals and design"; must be completely rewritten. unclear. 

line 380: "4.10. Statistical analysis".  the statistical analysis needs to be better described. The authors performed a simple comparison of mean values (t-test), probably for small sample sizes it was preferable to use a 'non-parametric t-test', e.g. the Mann-Whitney test.

References: the year of publication must be written in bold letters.